# Ways of Long-Term Survival of Hydrocarbon-Oxidizing Bacteria in a New Biocomposite Material—Silanol-Humate Gel

**DOI:** 10.3390/microorganisms11051133

**Published:** 2023-04-27

**Authors:** Yury A. Nikolaev, Elena V. Demkina, Ekaterina A. Ilicheva, Timur A. Kanapatskiy, Igor A. Borzenkov, Anna E. Ivanova, Ekaterina N. Tikhonova, Diyana S. Sokolova, Alexander O. Ruzhitsky, Galina I. El-Registan

**Affiliations:** The Federal State Institution “Federal Research Centre “Fundamentals of Biotechnology” of the Russian Academy of Sciences” (Research Center of Biotechnology RAS), Leninsky Prospect 14, 119991 Moscow, Russia

**Keywords:** hydrocarbon-oxidizing bacteria, silanol-humate gels, long-term survival, stress resistance, ultrastructural cell organization

## Abstract

Immobilized bacterial cells are presently widely used in the development of bacterial preparations for the bioremediation of contaminated environmental objects. Oil hydrocarbons are among the most abundant pollutants. We have previously described a new biocomposite material containing hydrocarbon-oxidizing bacteria (HOB) embedded in silanol-humate gels (SHG) based on humates and aminopropyltriethoxysilane (APTES); high viable cell titer was maintained in this material for at least 12 months. The goal of the work was to describe the ways of long-term HOB survival in SHG and the relevant morphotypes using the techniques of microbiology, instrumental analytical chemistry and biochemistry, and electron microscopy. Bacteria surviving in SHG were characterized by: (1) capacity for rapid reactivation (growth and hydrocarbon oxidation) in fresh medium; (2) ability to synthesize surface-active compounds, which was not observed in the cultures stored without SHG); (3) elevated stress resistance (ability to grow at high Cu^2+^ and NaCl concentrations); (4) physiological heterogeneity of the populations, which contained the stationary hypometabolic cells, cystlike anabiotic dormant forms (DF), and ultrasmall cells; (5) occurrence of piles in many cells, which were probably used to exchange genetic material; (6) modification of the phase variants spectrum in the population growing after long-term storage in SHG; and (7) oxidation of ethanol and acetate by HOB populations stored in SHG. The combination of the physiological and cytomorphological properties of the cells surviving in SHG for long periods may indicate a new type of long-term bacterial survival, i.e., in a hypometabolic state.

## 1. Introduction

The remediation of environmental systems contaminated with liquid pollutants such as hydrocarbons, pharmaceutical waste, etc., has become an urgent issue during the last decades; of these, oil spills are the most large-scale problem [1,2]. Remediation usually includes the application of hydrocarbon-oxidizing bacteria (HOB) [3,4]. In the case of oil contamination, large-scale bioremediation required the development of efficient HOB-based preparations, as well as addressing issues such as the long-term preservation of high viable HOB titer during storage, transportation, and the application of the preparations and their improved introduction into contaminated soil biotopes [4,5,6]. HOB storage in an immobilized state on the surface of sorbents or incorporated into polymer gels is a promising approach to these problems [7,8].

Abundant information is now available concerning the efficiency of immobilized microbial cells as target-oriented biocatalysts; attention is focused on the preservation and enhancement of the desired enzymatic activity of immobilized microorganisms during their industrial application [9,10].

Another important issue is the successful introduction of new microorganisms into environmental microbial communities. The success of this introduction may be limited by the toxicity of the contaminated biotopes. Oil contamination is usually accompanied by the stress pressure of high heavy metal concentration, salinity, etc. [11,12,13]. These problems are usually addressed by a selection of various HOB genera, species, and strains, which are relatively resistant to these stress factors, e.g., by using the relatively stress-resistant hydrocarbon-oxidizing *Rhodococcus* strains [5].

Efficient bacterial preparations for the bioremediation of oil-contaminated ecosystems also require the following prerequisites: the preservation of high numbers of viable cells during long-term storage, transportation, and the application of the biopreparations; the ability of the population to be introduced into the native microbial community; and the preservation of the viability and targeted activity under the impact of high concentrations of heavy metals, salinity, etc.

We have previously shown that the immobilization of HOB cultures into the gels based on humates and (3-aminopropyl)triethoxysilane (APTES) resulted in long-term HOB survival (12 months and longer during storage at temperatures of ~20 °C [6,14]. The titer of surviving cells was up to 1000 times higher than in the control, and the ability of these preparations to oxidize hydrocarbons was higher compared to the control (planktonic cultures).

The aim of the present work was to determine the mechanisms of long-term HOB survival and of the relevant morphotypes for members of the genera *Rhodococcus*, *Acinetobacter*, and *Pseudomonas* embedded in new biocomposite materials based on silanol-humate gels.

## 2. Materials and Methods

Research subjects were hydrocarbon-oxidizing bacteria from the collection of the Laboratory of Petroleum Microbiology, Federal Research Center for Biotechnology, Russian Academy of Sciences: gram-negative bacteria *Acinetobacter seifertii* WS1, *Pseudomonas extremoaustralis* WS1, and *P. aeruginosa* 01S4.8.1 and a gram-positive bacterium *Rhodococcus qingshengii* 367-6 (formerly *R. erytropolis*). *P. aeruginosa* is known as a hazardous species, so it was used by us only as a model, and it is not recommended for introduction into natural ecosystems.

The cultures were grown in LB medium (Miller, Luria-Bertani, Sigma-Aldrich) in 250-mL flasks with 50 mL of the medium at 28–30 °C on an orbital shaker (100 rpm) to the stationary growth phase (24–40 h).

The new biocomposite material, silanol-humate gel with immobilized bacterial cells (SHG), was produced using the Powhumus sodium-potassium humate (Humintech, Germany) and the organosilane (3-aminopropyl)triethoxysilane (APTES) (Dia-m, Russia) according to the previously described procedure with certain modifications [14,15,16]. Humate (1.5 g) was dissolved in 10 mL of distilled water under vigorous agitation on a magnetic stirrer, and the undissolved fraction was separated by centrifugation (5000× *g*, 5 min). The supernatant was supplemented with 0.5 mL APTES under vigorous agitation on a magnetic stirrer and titrated to pH 6–7 with diluted acetic acid. The stationary-phase bacterial suspension (5 mL) was then added. The mixture was left at room temperature for 2–12 h to achieve gelation of the reaction mixture. As a result, bacterial cells were incorporated in the matrix of a three-dimensional silsesquioxane structure with attached humate polyanions. The gel containing bacterial cells (further on designated SHG) was homogeneous, possessed moderate strength properties (did not leak out of overturned test tubes), and preserved its properties for several months. SHG contained acetate (1.3 g/L) and ethanol (2 g/L), which were produced in the course of its formation. Importantly, the gel dissolved upon the addition of water or intense agitation (Vortex, 3 min).

The viability of the HOB cells was determined as the number of colony-forming units (CFU/mL) obtained by plating 5-µL of decimal dilutions of the cultures or of the liquefied SHG preparations (in 5 to 7 replicates) on solid LB medium.

The metabolic activity of the HOB was assessed by their respiratory activity, which was determined by two methods. First, exogenous respiration was determined by CO_2_ accumulation in the headspace of hermetically sealed 120-mL vials with 10 mL of the Raymond salt medium containing the following (g/L): CaCl_2_·6H_2_O–0.01; MnSO_4_·5H_2_O–0.02; FeSO_4_·7H_2_O–0.01; Na_2_HPO_4_–1.5; KH_2_PO_4_–1.0; MgSO_4_·7H_2_O–0.2; NH_4_NO_3_–1.0; NaCl–5.0; pH 6.8–7.2. Oil from the Cheremukhov deposit (Tatatstan) with the density of 0.9 g/cm^3^ was used as the carbon source (2% *v/v*). The medium was inoculated with 200 µL of the stationary-phase planktonic culture or with 600 µL of the liquefied (Vortex, 3 min) SHG preparation, since during SHG formation, the initial culture was diluted threefold (see above). The vials were incubated for 10 days at 30 °C with agitation (100 rpm). The volume of the medium and the amount of the substrate were adjusted so that oxygen did not limit oil oxidation in the closed volume. Air samples (0.4 mL) were periodically collected from the vials, and CO_2_ concentration was determined on a Chromatek-Krystal-5000 chromatograph (Chromatek, Russia). To calculate CO_2_ emission from the amount formed, the CO_2_ concentration in the control (without oil) was subtracted. Alternatively, respiration was assayed according to CO_2_ accumulation in the culture liquid. For this purpose, 2-mL samples were taken hourly from cell suspensions (culture aliquots or SHG preparations) resuspended in Raymond medium without carbon sources. The samples were transferred into 18-mL Hungate tubes, sealed hermetically, and supplemented with 0.1 mL of 20% phosphoric acid. CO_2_ concentration in the gas phase was determined as described above.

The ability of bacteria to degrade individual hydrocarbons or oil in liquid Raymond medium was determined according to CO_2_ accumulation in the gas phase in sealed vials, as was described above.

To determinate ethanol and acetate concentrations in SHG preparations, the liquefied SHG were dissolved (1:1 *v/v*) in 1% phosphoric acid (Fluka lot #STBF 2313V). The dissolved sample was mixed with an equal volume of *n*-octanol (Pancreac lot #0001989683). After thorough stirring, the mixture was centrifuged (3000× *g*, 3 min) to achieve phase separation. The amounts of acetate and ethanol in the upper phase were determined by gas-liquid chromatography (Shimadzu GC 2010+ with a GCMS QP 2010 Ultra mass detector, Shimadzu).

For the construction of the calibration curves for ethanol and acetate, a liquefied sample (SHG without cells) was dissolved in 1% phosphoric acid and supplemented with known amounts of these compounds.

The conditions for gas chromatography were as follows: carrier gas, helium; flow rate, 0.88 mL/min with 1:200 flow separation (flow linear rate was 34 cm/s); and the MDN-5 column (bound methyl silicone) 30 m × 0.25 mm × 0.25 µm. The temperature program was as follows: detector, 200 °C; interface, 205 °C; thermostat: 45 °C for 3 min, then to 50 °C at 5 °C/min and to 260 °C at 30 °C/min, and then isothermal mode for 1 min. Mass spectra were recorded at minutes 1–7 within the *m*/*z* range of 14–150.

The biosurfactants amount was assessed by their activity: changes in the surface tension (ST), interfacial tension (IT), and emulsification index (E_24_) [17]. ST was determined by the semistatic ring detachment method on a Surface Tensiomat 21 semiautomatic tensiometer (Cole-Parmer, Vernon Hills, IL, United States) with a platinum-iridium ring. IT was measured at the liquid/hexadecane phase boundary. To establish the phase boundary, the samples were incubated for 30 min. All measurements were carried out at 25 °C.

The emulsification index (E_24_) was determined according to Zajic et al. [18] with certain modifications. Hexadecane (1.0 mL) was supplemented with an equal volume of the studied bacterial culture or SHG sample. The mixture was shaken for 3 min, and the volume and stability of the formed emulsion were determined after incubation for 24 h at 20 °C. The index was expressed as the percentage of emulsion volume in the total volume of the mixture.

Ultrastructural cell organization was studied by the transmission electron microscopy of ultrathin samples.

Gel samples were fixed with 2.5% glutaraldehyde solution in cacodylate buffer (0.05 M sodium cacodylate; pH 7.0–7.5) at 4 °C for 24 h. They were then washed three times with 5 mL of the same buffer, and fixed with 1% OsO_4_ in 0.7% solution of ruthenium red in cacodylate buffer for 1.5 h at 4 °C. After fixation, the samples were embedded in 2% agar and incubated in 3% uranyl acetate in 30% ethanol for 4 h and then in 70% ethanol fro 12 h at 4 °C. The material was dehydrated in 96% ethanol (twice for 15 min) and then in absolute acetone (3 times for 10 min). The material was then embedded in Epon-812 (Epoxy Embedding Medium Epon^®^ 812, Sigma-Aldrich, Burlington, MA, USA): in the 1:1 mixture with acetone for 1 h and in 2:1 mixture for 1 h. Polymerization was carried out for 24 h at 37 °C and then for 24 h at 60 °C. Ultrathin sections obtained using an LKB-III microtome (LKB, Sweden) were contrasted with 3% aqueous uranyl acetate (30 min) and with 4% aqueous lead citrate (30 min).

The preparations were investigated under a JEM 100CXP electron microscope (JEOL, Tokyo, Japan) at an accelerating voltage of 80 kV and working magnification of 5000–50,000×. The results were documented using the Morada G2 digital system.

The phenotypic phase variants of the HOB cultures were determined according to the colony morphology on LB agar inoculated with the decimal dilutions (500 µL) of bacterial cultures (stationary or stored for long terms), as well as according to their ability to grow under extreme conditions (at subinhibitory concentrations of NaCl and Cu^2+^).

Data processing: statistical analysis was performed using the standard methods implemented in Microsoft Excel XP. The data groups were considered uniform if the standard deviation did not exceed 10%. The differences between groups were considered significant at *p* > 0.95. Standard errors were calculated and are present with all values in the figures as data deviations.

## 3. Results

### 3.1. Physiological Characteristics of SHG-Immobilized Cells

#### 3.1.1. Preservation of Cell Viability

SHG-embedded cells of HOB of diverse taxonomic positions varied in the preservation of their viability, as was evident from their ability to form colonies on solid media. Thus, the CFU titer of *P. aeruginosa* after 12 months of storage was ~100% of the initial value, i.e., it was 100 times higher than in the control variant, the planktonic culture stored for 12 months [6]. The survival of SHG-immobilized *A. serfertii*, *P. extremoaustralis* and *R. qingshengii* was an order of magnitude higher than in the control (without the gel).

Microbial cells, as a rule, exhibit long-term viability either when growing (e.g., in a flow culture) or in a state of anabiosis (metabolic dormancy). To determine the state of the SHG-immobilized HOB cells, some of their characteristics were investigated.

#### 3.1.2. Respiration of the HOB Cultures on Oil Paraffins

SHG-immobilized *A. seifertii* cells, which were used as inocula, exhibited high respiratory activity with oil as the exogenous substrate. The rates of cell respiration stimulated by oil paraffins in the cultures inoculated with aliquots of 2-day and 4-month *A. seifertii* cultures are shown on Figure 1. Cell reactivation in the cultures inoculated with the material from SHG-immobilized samples was the highest, with the shortest lag phase and the quickest development of intense respiration. The SHG variant also exhibited the highest rate of CO_2_ accumulation after 250 h of the experiment (315 ppb CO_2_/h).

#### 3.1.3. Endogenous Respiration of HOB Cells

The metabolic activity of SHG-immobilized *A. seifertii* cells after 4-month storage was determined by measuring their endogenous respiratory activity, i.e., using the substrates present in the incubation medium. The initiation of endogenous respiration was expected to be initiated by the dilution of the cultures with fresh Raymond medium, heating to 30 °C, and oxygen supply due to shaking.

In this experiment, a more sensitive approach to measuring the respiration rate by the rate of CO_2_ accumulation in the liquid phase was used. The results, calculated as µg CO_2_/mL suspension/h, are presented in Figure 2.

The cells stored in the gel for 4 months exhibited higher rates of endogenous respiration than both controls (planktonic cultures of 2-day and 4-month cultures) (Figure 2). It should be noted that the respiration rate was measured for the cells suspended in salt medium, heated to 30 °C, and supplied with oxygen. The respiration rate of *A. seifertii* cells surviving in the gel was 1.8 times higher than that for the 4-month planktonic culture. Unexpectedly, immobilized cells exhibited the rate of endogenous respiration 1.3 times higher than that of a fresh (48 h) planktonic culture. Thus, the cells surviving in SHG for 4 months or longer were not in a state of metabolic dormancy.

The more active respiration of SHG-immobilized cells compared to those of the stationary cultures may be due to two reasons. First, the cells grown in the rich LB medium were not adapted to paraffin utilization, unlike the cells stored in SHG. Second, the high respiration rate in the absence of intense growth indicated that this respiration was not productive, which is not contradicted by the data on intense respiration with oil (Figure 1).

#### 3.1.4. Detergent Formation

The synthesis of the surfactants typical of hydrocarbon-oxidizing bacteria by *A. seifertii* was a direct confirmation that SHG-immobilized cells were not in a state of metabolic dormancy.

The surface and interfacial tension (ST and IT) and the emulsification index (E_24_) were determined for liquefied samples of SHG with *A. seifertii* cells, which have been stored for 7 months. The gels were liquefied prior to analysis by active mixing (Vortex, 3 min) or by diluting 1:20 with tap water. Sterile tap water and liquefied SHG without bacterial cells were used as the controls. The results presented in Table 1 show the presence of surfactants according to all three parameters: both the ST and IT values were lower than in the control, while the water-phase emulsion in hexane was stabilized (E_24_ was higher than in the control). Thus, SHG-immobilized HOB cells retained their metabolic activity and were able to produce surfactants, even in the absence of oil hydrocarbons. Surfactant synthesis in our experiments was probably a response of the cells to starvation stress or to the proximity of the polymers present in SHG.

#### 3.1.5. Dynamics of Acetate and Ethanol Content in SHG Samples with HOB during Long-Term Storage

The consumption by the HOB cells of acetate and ethanol formed during SHG preparation was a direct indication of the active metabolism of SHG-immobilized cells.

The concentrations of acetate, ethanol, and CO_2_ in the liquid phase of the SHG-immobilized HOB preparations are presented in Table 2. While the concentrations of acetate and ethanol were determined quantitatively and are presented in mM, CO_2_ concentration is expressed in conventional units, as percentages of its maximal content in experimental SHG samples with *A. seifertii* after 18 months of storage. Quantitative CO_2_ determination was not our goal at this stage, and the experimental design was aimed at the indication of the presence or absence of CO_2_ in the medium.

Acetate was consumed completely by all three HOB strains investigated, and its content decreased during long-term storage from >200 to 2–5 mM. Ethanol was actively consumed only by SHG-immobilized cells of *R. qingshengii* and *P. extremoaustralis*, while its consumption by *A. seifertii* was insignificant.

*R. qingshengii* consumed ethanol less actively than acetate, so that 60% of acetate and only 14% of ethanol were consumed after 6 months; however, both substrates were consumed almost completely after 9 months (99 and 98%, respectively). In all variants, CO_2_ was detected in the medium, unlike the control (cell-free SHG).

Thus, after long-term storage (up to 18 months), SHG-immobilized hydrocarbon-oxidizing bacteria retained their ability to oxidize carbon sources, and were therefore in a metabolically active (not dormant) state.

#### 3.1.6. Phase Variations of the *P. extremaustralis* Populations

The phase variation spectrum of the populations emerging from surviving cells plated on solid media may be used as an additional parameter of the physiological state of long-stored cells. The populations emerging from the dormant, cyst-like cells exhibited the spectrum of colony morphology variants different from the one predominant in the parent population of actively growing cells [18,19].

The «dominant (large)» <=> «minor (small)» model of colony morphology transitions for *P. aeruginosa* (Figure 3, insert) was used to show that the phase variation spectrum of the growth phases and the early stationary phase consisted of predominant large colonies (90%) and of minor small colonies (10%). The natural development of dormant forms (DF) and their subsequent germination resulted in another character of the population, in which each variant constituted 50% (Figure 3).

After the 4-month storage of SHG-immobilized samples, the phase variation spectrum of the emerging population was close to that of a growing culture, with the shares of the dominant, large-colony variant and the small-colony one being 85 and 15%, respectively. These results confirmed that the bacteria surviving in SHG were mainly represented by vegetative cells. However, the decreased relative abundance of the dominant colony type indicated the presence of a small number of dormant forms.

Since the populations emerging from DF are known to produce a broader spectrum of variants, differing, apart from colony morphology, in their physiological and biochemical characteristics [19,20], we investigated the ability of the SHG-stored cells to grow under extreme conditions and on the substrates unusual for HOB.

#### 3.1.7. Ability of SHG-Stored HOB Cells to Oxidize Oil in Liquid Media under Stress Conditions

The ability of *A. seifertii* to grow in liquid medium with oil in the presence of 10^−3^ M Cu^2+^ or 3% NaCl was studied for the media inoculated with 2-day and 5-month stationary cultures (controls) and SHG-immobilized material after 5-month storage (Figure 4). Hydrocarbon oxidation was monitored by CO_2_ formation.

While in both control variants, the metabolic activity of *A. seifertii* was observed after a prolonged lag phase (100–170 h of incubation), inoculation with SHG-immobilized cells resulted in almost immediate paraffin oxidation, so that the amount of the substrate oxidized after 200 h of incubation in the presence of Cu^2+^ was 1.5 and 6 times higher than in the control variants of 2-day and 5-month planktonic cultures (CO_2_ concentrations were 6, 4, and 1%, respectively) (Figure 4a). The high metabolic activity of SHG-inoculated cultures may be explained either by Cu^2+^ sorption properties by the SHG components (~6% *w/w* in the medium), and therefore the less pronounced effect of these ions, or by the emergence in long-stored SHG of a variant resistant to stress factors, including Cu^2+^. It should be noted that SHG was specifically designed for the sorption of various toxic agents, including metal ions [14,15,16].

In the presence of another stress factor, 3% NaCl (the concentration was determined to be subinhibitory in preliminary experiments), the behavior of *A. seifertii* cells was different (Figure 4b). The cells of the 2-day stationary culture did not adapt to high salinity during 200 h of the experiment, and no CO_2_ accumulation occurred. Inoculation with a 5-month liquid culture containing *A. seifertii* dormant forms resulted in an accumulation of up to 0.04% CO_2_ after 200 h. This was caused by the activity of the variant developed during DF germination. The variants inoculated with SHG-immobilized bacteria stored for 5 months exhibited less active paraffin oxidation, with a CO_2_ accumulation of up to 0.01%. Since the SHG polymers do not sorb NaCl (unlike Cu^2+^), the only plausible reason for growth in the presence of NaCl was the emergence of a halotolerant variant (phenotype). The development of DF in SHG-immobilized material is reported below.

Thus, our data confirmed the presence of dormant, non-metabolizing forms in *A. seifertii* populations and the realization of its adaptive potential via the emergence of the variants resistant to stress factors (heavy metals and salinity).

#### 3.1.8. Stress Resistance of HOB Phase Variants Grown on Solid Media

The above conclusion concerning HOB survival in SHG samples as stress-resistant variants was confirmed by a series of experiments on HOB phase variability on solid media. Surface cultivation was preferred, since it makes it possible to determine both the number of viable cells and the characteristics of colony morphology. Stationary cells (proliferation rest) and DF (both proliferation and metabolic rest) were used as inocula. In the latter case, the phase variant spectrum of the population broadened. Preliminary experiments revealed the subinhibitory concentrations of the stress agents: NaCl concentrations 3.6–9%; and CuSO_4_·5H_2_O, 2 × 10^−3^ M, 10^−3^ M, and 10^−4^ M. The experiments were carried out for 30 days at room temperature. The results are summarized in Table 3 and Table 4.

The highest NaCl concentration at which *A.seifertii* was able to grow was 3%. The DF-derived population had the relative abundance of the salt-resistant variant (forming small, slimy, pinkish colonies), which was 15 times higher than in the population derived from the stationary cells (Table 3). *R. qingshengii* could grow at salt concentrations of up to 9%, and the relative abundance of salt-resistant cells (forming small transparent colonies, d < 0.2 mm) in the population obtained from DF was two orders of magnitude higher than the stationary cells that were used as inoculum (Table 3).

A similar situation was observed with another stress factor, copper sulfate CuSO_4_. The population obtained from DF contained 3–20 times more copper-resistant cells than the population obtained from the stationary cells (Table 4).

These results confirmed the previously determined property of bacterial cyst-like DF to develop in a new growth cycle as populations of different phenotypic variants.

#### 3.1.9. Ultrastructural Characteristics of the HOB in Long-Stored SHG Samples

The ultrastructural organization of the HOB surviving in the gel for long periods were determined in SHG samples of different density. The gels containing 3% *w/w* of humate content of the gelating agent, APTES, had a soft consistency of thick cream and could leak out of overturned test tubes. The gels with 5% organosilane were considered dense; they did not leak out of overturned tubes. The samples were investigated after gel liquefaction by vigorous shaking (Vortex, 3 min). The results are presented in Figure 5, Figure 6, Figure 7, Figure 8 and Figure 9.

One day after the incorporation of the stationary HOB cells in the gel, their structure was typical of the stationary cells in liquid culture (Figure 5a,c), with uniform cytoplasm of moderate density. Some cells were irregularly shaped, with constrictions, indicating the incomplete fission process (Figure 5b,d). Thus, the incorporation of the cells in SHG exhibited no pronounced negative effect.

After 3 months of storage, all preparations of SHG-immobilized HOB were characterized by the morphological heterogeneity of the cells (Figure 6 and Figure 7).

*R. qingshengii* cells stored in a soft gel for 3 months were mainly stationary cells with compacted biocrystallized nucleoid (Figure 6a) and dormant forms of two morphotypes: with the cytoplasm of moderate electron density and a moderately thickened cell wall (Figure 6b) or with the uniformly electron-dense cytoplasm and a thick cell wall (Figure 6c). A small portion of cells were connected with pili up to 5 µm long and up to 150 nm wide (Figure 6d). Such pili were not revealed in the hard gel. Interestingly, some cells possessed membrane projections, which were converted into membrane vesicles (Figure 6d,e). Such structures are typical of the biofilm state of bacteria [21,22,23,24,25]. Some stationary cells possessed a pronounced capsule (Figure 6f), which is also characteristic of the biofilm phenotype. No capsules were found around the metabolically inert DF (Figure 6b,c).

The composition of the protein and carbohydrate pool changes in immobilized microorganisms; some of them are converted to the “capsular” microenvironment [26]. The formation of such capsules was observed after the 3-month storage of SHG-immobilized *R. qingshengii* (Figure 6f) and *A. seifertii* (Figure 7a). Lysed cells constituted a minority in 3-month *R. qingshengii* preparations (less than 5% of the population).

The samples of SHG-immobilized *A. seifertii* after 3 months of storage presented a different picture. Unlike *R. qingshengii*, a significant part of the cells were lysed and formed large transparent cells (sheaths up to 6 µm in size) (Figure 7c,d). A part of the population was represented by the stationary cells of regular size (Figure 7a), while another part consisted of dormant cells belonging to two morphotypes: with the cytoplasm and envelope of moderate electron density and a well-visualized, compacted nucleoid (Figure 7b); and DF with the cytoplasm of high electron density (Figure 7c). Unusual pairs of divided, but not separated cells were observed, with one of them (the larger) autolyzed and another (4–5 times smaller) having the ultrastructure typical of old stationary cells (Figure 7c,d). While the increased size of adhered cells was described earlier [27], their autolysis with the survival of the separated cell (4–5 time smaller) has not been previously reported. A minor part of the population was represented by very small viable cells, not exceeding 260 nm in width (Figure 7f).

The populations of *A. seifertii* cells stored in the soft SHG were characterized by the massive formation of pili connecting the individual cells (Figure 7e). These were probably F-pili, responsible for the horizontal transfer of genetic material during transformation or transduction. The enhanced horizontal transfer of genetic determinants has been known to occur in the populations of immobilized cells [28,29], although we are not aware of reports on the visualization of the cell structures responsible for this process.

Interestingly, the formation of F-pili was not observed in *A. seifertii* populations stored in the hard gel. In this case, the cells were represented by the stationary cells (Figure 8a) and two types of dormant cells with multilayered envelopes (Figure 8b) or with a thickened dense, single-layer envelope and electron-dense cytoplasm (Figure 8c).

The morphological structures observed in the population of another gram-negative bacterium, *P. extremoaustralis*, stored for 3 months in soft SHG, were similar to those exhibited by *A. seifertii*: pili, large lysed cells (Figure 9a), including those connected by constrictions with small electron-dense cells (Figure 9b,c). Unlike *A. seifertii*, numerous completely lysed cells (sheaths) were observed in the pseudomonad population (Figure 9a–c).

## 4. Discussion

Under natural conditions, including those of soil ecosystems, microorganisms are attached to solid surfaces, and usually exhibit the biofilm phenotype. In biotechnology and medicine, the biofilm phenotype is studied mainly as a form of development of pathogenic bacteria and a cause for biological damage to materials and structures. Abundant literature is available concerning the investigation of bacteria immobilized on solid carriers or incorporated into the spatial structure of a carrier and to their applications as catalysts of required enzymatic reactions [9,10,30].

We have previously developed a new biocomposite material based on HOB of various genera embedded into an organosilane gel modified with humates (SHG) [8,14]. The material was intended for the long-term storage of bacteria (3 to 15 months), rather than for their application as biocatalysts. These gels are easily soluble in water and act as sorbents for heavy metals and other pollutants [15,16]; they contain humates acting as soil meliorants and antioxidants [31,32]. The chemical structure of SHG is similar to the chemical composition of soils. Importantly, due to the technology of its production, SHG contains energy sources, acetate, and ethanol (≈200 mM).

The incorporation of stationary-phase cells rather than dividing bacterial cells, which are, as a rule, used to obtain immobilized cell preparations [9,10,30], is another specific feature of the new biocomposite. The reasons for this are as follows. Unlike dividing cells, the stationary ones are considerably more resistant to the damaging agents and implement the stationary phase transcription programs (SOS-response, the OxyR and RpoS regulons, etc.), including the synthesis of the nucleoid-associated stationary phase protein, Dps [33,34]. In the stationary-phase cells, the Dps protein has three important functions. First, as a ferritin family protein, Dps protects the cell structures from free radicals produced via the Fenton reaction. Second, its co-crystallization with DNA due to physicochemical interactions is responsible for its compaction and reversible inhibition of transcription activity [35]. Third, their interaction with each other (Dps-Dps) and with DNA (DNA-Dps) produces a unique structure, a biocrystallized nucleoid, in which DNA, RNA, and the nucleoid proteins are protected from the relevant depolymerases and external stress agents [36]. Moreover, a stationary bacterial population contains a subpopulation of persister cells (~1%), which are formed in the course of cell differentiation [37,38] and, according to our hypothesis [39,40], are precursors of cyst-like anabiotic dormant forms (DF). The transition of a growing population to the stationary phase is always associated with the critical concentration of the density autoregulators of one out of two types: QS system autoinducers [41] or d_1_ factors (alkylresorcinols, etc.) [42].

After several days of the stationary phase, autolysis commences in liquid (planktonic) cultures, affecting ~99% of the ordinary cells, while ~1% of persister cells [38] remain intact and autolysis-resistant unlike other cells [43] and require the nutrients present in the autolysate to maintain their viability [44,45]. Autolysis is induced by a critical increase in the concentration of extracellular unsaturated fatty acids and is then enhanced by the hydrolases released from the already autolyzed cells [42].

The dynamics of death/survival for SHG-immobilized bacteria were quite different from non-immobilized cells [6,14]. The titer of the HOB cells surviving for several months was several orders of magnitude higher than in the control planktonic cultures. This unnaturally long preservation of the viable cell titer (CFU) in the absence of nutrient sources N and P has not been observed previously and requires an explanation.

The results presented in this work indicate the following possible causes for the unusually long survival of SHG-immobilized bacterial cells.

The preservation of cell precipitation to the bottom of the cultivation vessel and their fixed localization inside the gel cells, together with the sorption properties of SHG and the threefold dilution of the culture during the preparation of the gel, prevent the accumulation of autolysis autoinducers in critical concentrations, which occurs in the cell precipitate of planktonic cultures. This may be the reason for the absence of cell autolysis, which was confirmed by electron microscopy (Figure 5, Figure 6, Figure 7, Figure 8 and Figure 9).

However, even intact cells may perish due to the oxidative stress accompanying every other stress, including starvation stress. This did not occur for several reasons.

Highly stress-resistant dormant forms, providing the long-term survival of the population, were present in the surviving populations. However, the share of DF in SHG did not exceed several per cent (TEM data), while the preservation of up to 100% of the population was observed in certain cases.

In the course of SHG preparation, the stationary bacterial cultures were diluted threefold, which decreased the critical concentration of the dormancy autoregulators and made it possible for some of the cells in the stationary population to resume their growth. This is in agreement with the fact that most SHG-immobilized HOB cells were in a physiologically active state rather than in a dormant one, as was confirmed by the data on endogenous respiration (Figure 2), biodetergent synthesis (Table 1), and the consumption of the energy substrates (acetate and/or ethanol) together with CO_2_ accumulation during long-term storage (Table 2). The cells predominating in the surviving population resulted from the secondary growth on the products of autolysis of the original inoculum. Importantly, the lysed cells were several times (3–5) larger than the intact viable cells (Figure 7), which supported the notion of the secondary growth of SHG-stored bacteria.

Although the extremely long-term functional activity and high operational stability of the cells immobilized in or on a carrier have been noted previously [9,10], this was always when the nutrient sources were present, e.g., oil hydrocarbon in the case of HOB [5]. While no external nutrient sources were used in our experiments, the cells survived without transitioning to a dormant state. In our opinion, the preservation of SHG-immobilized stationary cells in their intact state was supported by acetate and ethanol (200 mM), which were produced in the course of gel formation and acted as internal energy sources; their role was confirmed by their consumption in the course of SHG storage. The increased size of gel-immobilized cells was previously attributed to their growth [27,46]. Cell size decrease over the course of the long-term survival of gel-immobilized microorganisms has not been reported prior to the present work.

Finally, it should be noted that immobilization in SHG, as in any other gel, results in the transition of the cell metabolism to the one more resistant to stress agents, e.g., the biofilm one [46]. The populations embedded in SHG also contained the cells with the biofilm phenotype (albeit in small amounts), as was indicated by the presence of the pili, membrane projections, and vesicles, as well as the capsules, which were observed in thin sections (Figure 6e,f).

Analysis of the ultrastructural organization of the HOB surviving in SHG for long periods revealed some interesting patterns. First, the long-time surviving bacterial populations were morphologically heterogeneous (Figure 6, Figure 7, Figure 8 and Figure 9) and included intact stationary cells and dormant forms (as was evident from their characteristic morphology and ultrastructure), as well as autolyzed cells. Intact cells were three to five times smaller than the autolyzed ones (~1 µm and several µm, respectively), and some of the surviving cells had the size typical of ultramicrobacteria (below 300 nm), which has not been reported previously. These data confirm the previously stated suggestion of partial autolysis and secondary growth of a part of the SHG-immobilized population in the absence of a balance nutrient medium. Second, the population of cyst-like DF was heterogeneous and included DF with multilayered or single-layer thickened envelopes (Figure 6 and Figure 8), as has been previously reported for DF of non-spore-forming bacteria of various taxa [18,47]. Third, the massive formation of pili (presumably type F) in the samples of SHG-immobilized HOB cells, which was more pronounced in the cases of *A. seifertii* and *P. extremoaustralis*, supports and confirms the previously shown higher ability of immobilized bacterial cells (compared to planktonic cultures) to carry out the exchange of genetic material [28,29]. Cell immobilization in SHG promotes the formation of fertile pili and provides for the preservation of their intact, undamaged state.

Survival as a new phase variant and thus, the implementation of the genetically determined adaptive potential of the populations, played an important part in the long-term survival of the HOB cells upon a drastic change of conditions (transfer from LB to SHG, changed medium density, decreased O_2_ concentration, etc.) [48,49,50]. We have repeatedly observed this broadening of the phase variation potential for cyst-like dormant forms plated on solid media [18,19,47,51]. The phase variants of the studied HOB species revealed after storage in SHG differed from the dominant variants used for inoculation in their colony morphology and physiological and biochemical characteristics. This intrapopulational biodiversity uses the adaptive potential of the population, so that the variant most adapted to specific conditions becomes the dominant phenotype. In this work, the populations of HOB *A. seifertii* and *R. qingshengii* originating from DF and of the cells stored in SHG for a long time were shown to develop as the phenotypes resistant to extreme environmental loads: salinity and high content of heavy metals (Cu^2+^) (Table 3 and Table 4). The variants adapted to these stress loads may be considered the ecological ones.

Considering the changes in the type of metabolism and morphology of the HOB cells transferred into new conditions (from full-strength, liquid LB medium to a denser environment devoid of nutrient sources, and probably with a lower O_2_ content), the ecological and biochemical/metabolic aspects of the population deserve attention.

Two major microbial strategies according to the criteria of reproduction and survival/death are known in population ecology [52,53,54]. The r-strategy implies the rapid reproduction of organisms (cells) and therefore the rapid growth of the population, rapid exhaustion of the resources of the medium, and rapid death under unfavorable conditions, with the survival of a small subpopulation of the cells transferring to the dormant state. Such populations are characterized by pronounced uniformity, and their abundance is subject to significant fluctuations. In the case of the K-strategy, both cell growth and death are slow, while the intrapopulation diversity is higher.

Our results indicate a shift in the strategies of the studied HOB from r to K when the cells grown in rich LB medium are transferred into the gel medium devoid of nutrient sources. Cell death is suppressed under such conditions, and the intrapopulational diversity and adaptation increase due to the emergence of the new phenotypes during secondary growth.

Our observations supplement the r-K selection theory with the possibility of shifting strategies within a single population at different stages of its development. The possibility of such variations may also be of applied importance for the efficient control of the growth and death dynamics and the properties of microbial populations.

The biochemical/metabolic aspect of the long-term survival of aerobic bacteria under starvation conditions in SHG in a metabolically active state (with endogenous respiration and biosurfactant synthesis) implies the utilization of the energy sources available in the gel (acetate and ethanol) and coping with the oxidative stress developing under such conditions. 

Reactive oxygen species (ROS) are always formed in the respiratory chain and cause the undesirable oxidation of all biopolymers and metabolites, nucleic acids, proteins, and lipids. Various mechanisms for adaptation at the biochemical level have been described for aging cells under conditions of long-term survival. The main ones are the maintenance of ATP content/synthesis at the sufficient, albeit decreased, level [55,56], maintaining the level of reductive equivalents, NADH, NADPH, etc., and the activation of the stress and stationary regulons [56].

Switching to fermentative metabolism, resulting in decreased ROS production and ATP synthesis via substrate-level phosphorylation, has been described [57].

Importantly, the variants capable of utilizing amino acids and aryl glucosides from autolysates were found to emerge in *E. coli* populations undergoing long-term starvation [58].

The cultures in the state of long-term starvation were shown to exhibit new phenotypes, GASP and CASP, growing well and constantly active in the stationary phase, respectively, which implemented different genetic programs of the stationary phase [56]. Similar changes probably occurred in the case of long-term survival in SHG. However, the relative abundance of these variants did not exceed 1% of the total population, while the abundance of the cells surviving in SHG was at least an order of magnitude higher.

Pseudomonads were shown to cope with the oxidative stress by reconfiguring their metabolism, increasing the concentration of the antioxidant NADPH and decreasing that of the prooxidant NADH (increased level of NAD kinase and decreased level of NADP phosphatase), as well as suppressing the activity of the TCA cycle [59].

These mechanisms have been reported for surviving cells, irrespective of their share in the population. Similar mechanisms of coping with oxidative stress may be expected to operate in the case of HOB survival in SHG.

## 5. Conclusions

Thus, the following causes are responsible for the extremely long-term survival of SHG-immobilized HOB cells. (1) The HOB populations intended for storage were in the stationary growth phase and contained two cell phenotypes: (a) a small (~1%) subpopulation of persister cells capable of maturation to DF of reversion to growth and (b) metabolically active, although proliferatively resting stationary cells, also capable of reversion to growth in a liquid medium, of lysis (in the presence of inducers, unsaturated fatty acids, or of transition to a new state of old, hypometabolic cells. (2) Unique conditions were established in SHG: (a) embedded cells were spatially disconnected and diluted (the concentration of unsaturated fatty acids was decreased threefold), which hindered autolysis and extended the release of the autolysate as a nutrient source for secondary bacterial growth; (b) energy sources present in the gel (acetate and ethanol) provided the preservation of the cells in their intact state during long-term starvation (5–15 months; (c) density/viscosity of the medium (due to the presence of the gel) simulated the physical parameters of the biofilm environment, inducing the transition of the cells to the biofilm phenotype. (3) The antioxidant properties of humates present in the gel promoted cell protection against the oxidative stress.

As the result, the SHG-immobilized population was heterogeneous with the majority being metabolically active, not dividing cells, which were preserved in their intact and viable state for 15 months or longer.

The combination of physiological and ultrastructural properties of the cells surviving in SHG may be interpreted as a new type of bacterial survival, i.e., in a hypometabolic state. This suggestion will be tested in our further research on the molecular genetic and physiological characteristics of surviving cells.

## Figures and Tables

**Figure 1 microorganisms-11-01133-f001:**
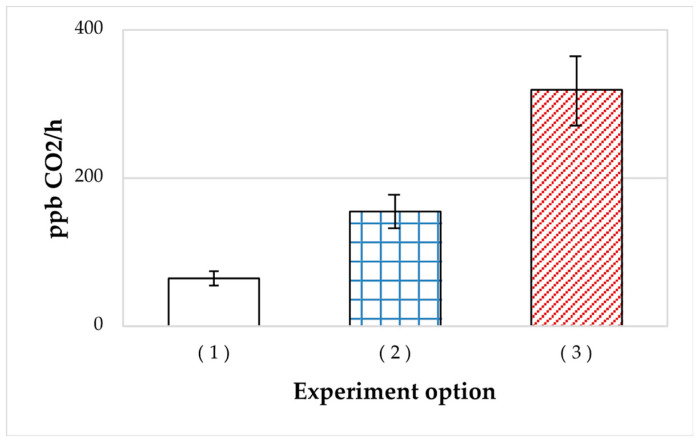
Rates of CO_2_ accumulation by *A. seifertii* cultures grown in Raymond medium with oil upon inoculation with: (1) liquid culture stored for 4 months; (2) the 48-h stationary liquid culture; (3) liquefied SHG stored for 4 months. Error bars are standard deviations.

**Figure 2 microorganisms-11-01133-f002:**
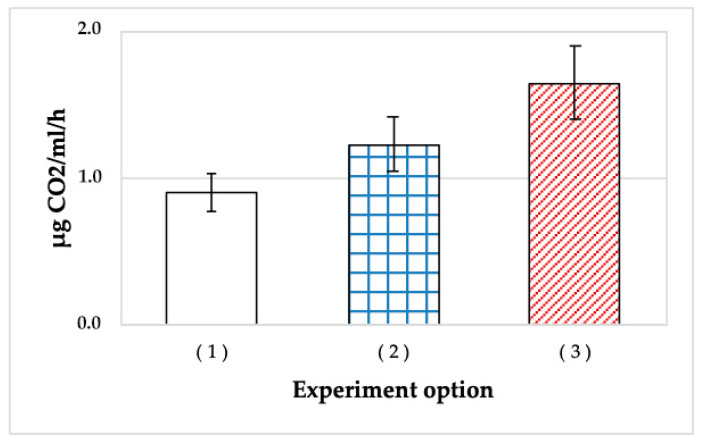
Rates of CO_2_ accumulation in the liquid medium by *A. seifertii* cultures of different age: (1) the culture stored for 2 months; (2) the 48-h stationary culture; (3) the liquefied SHG stored for 4 months. Error bars are standard deviations.

**Figure 3 microorganisms-11-01133-f003:**
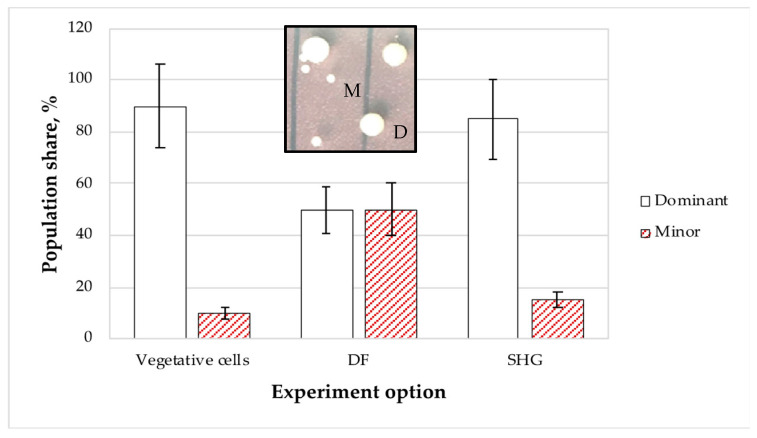
Phase variation spectra of the *P. aeruginosa* populations emerging from: vegetative cells, dormant cells (DF), and after storage in SHG. The insert shows the colony morphotypes: D–dominant, with large colonies and M–minor, with small colonies. Error bars are standard deviations.

**Figure 4 microorganisms-11-01133-f004:**
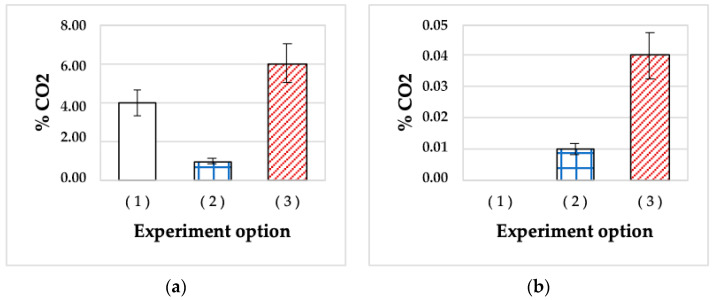
CO_2_ accumulation, % in *A. seifertii* cultures grown for 200 h in hermetically sealed vials on Raymond medium with oil and CuSO_4_ (10^−3^ M): (**a**) or 3% NaCl; (**b**) upon inoculation with: (1) 2-day planktonic culture; (2) 5-month planktonic culture; (3) 5-month SHG preparation. Error bars are standard deviations.

**Figure 5 microorganisms-11-01133-f005:**
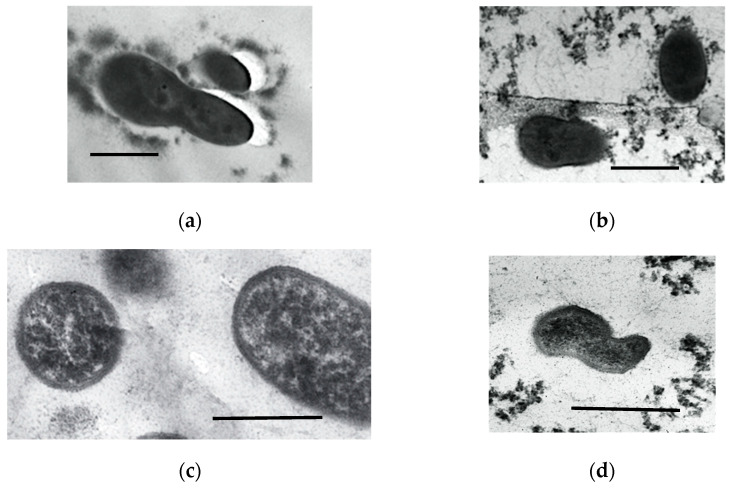
Ultrathin sections of the cells of *R. qingshengii* (**a,b**) and *A.seifertii* (**c**,**d**): (**a**,**c**) in the stationary growth phase; (**b**,**d**) in the SHG preparations stored for 1 day. Bar is 0.5 mkm.

**Figure 6 microorganisms-11-01133-f006:**
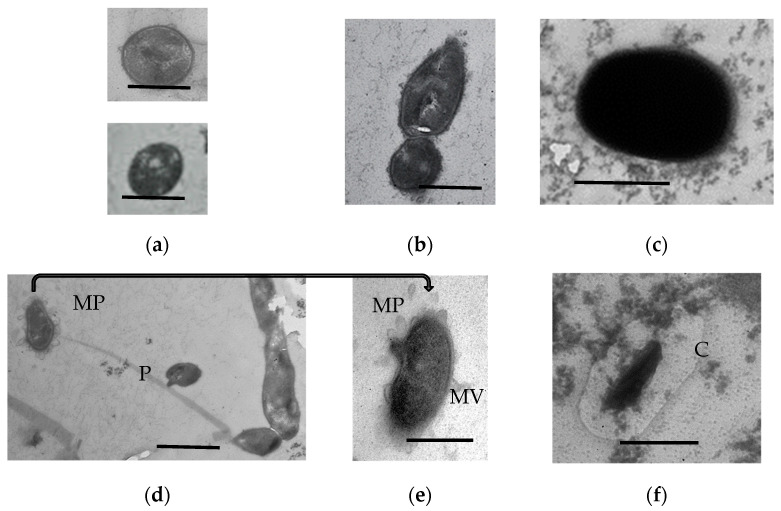
Ultrathin sections of *R. qingshengii* cells stored in the soft SHG for 3 months: (**a**) vegetative cells; (**b**) electron-dense DF with a compacted biocrystalline nucleoid; (**c**) DF with homogeneous electron-dense cytoplasm and a thick cell wall; (**d**,**e**) cells with pili (P) and abundant membrane projections (MP) transforming into membrane vesicles (MV); (**f**) vegetative cells with capsules (C). Bar is 0.5 mkm in (**a**–**c**,**e,f**) or 1 mkm in (**d**).

**Figure 7 microorganisms-11-01133-f007:**
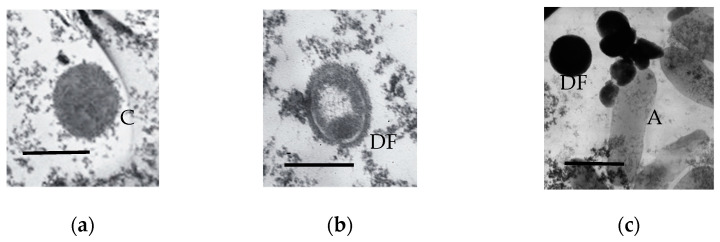
Ultrathin sections of *A. seifertii* cells stored in the soft SHG for 3 months: (**a**) vegetative cells; (**b**) DF with low electron density; (**c**) electron-dense DF and (**d**) autolyzed cells (A); (**e**) cells with pili (P); (**f**) micro-cells. C indicates a capsule. Bar is 0.5 mkm in b or 1 mkm in (**a**,**c**–**e**) or 0.25 mkm in (**f**).

**Figure 8 microorganisms-11-01133-f008:**
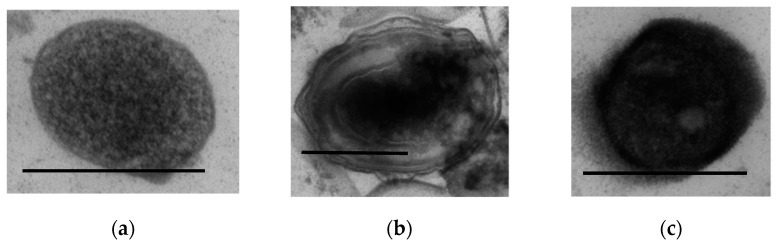
Ultrathin sections of *A. seifertii* cells observed after storage in hard SHG for 3 months: (**a**) cells of the stationary type; (**b**) DF with a multilayered envelope; (**c**) DF with a thickened, single-layer envelope and electron-dense cytoplasm. Bar is 0.5 mkm.

**Figure 9 microorganisms-11-01133-f009:**
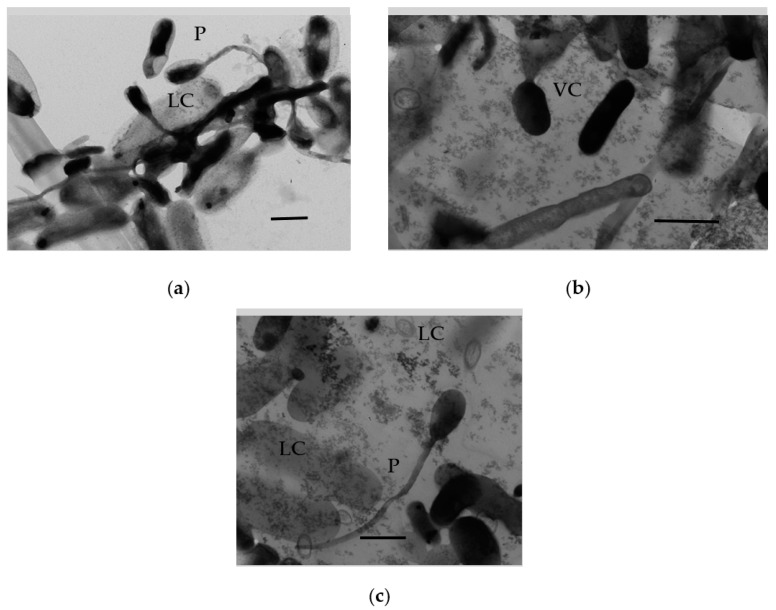
Ultrathin sections of *P. extremoaustralis* cells stored in the soft SHG for 3 months: pili (P), vegetative cells (VC), and lysed cells (LC). Bar is 1 mkm.

**Table 1 microorganisms-11-01133-t001:** Rheological and emulsifying characteristics of the liquefied *A. seifertii* cultures after 7-month storage in SHG.

	ST, mN/m	IT, mN/m	E_24_, %
Control (SHG without cells)	58.0	28.9	0
*A. seifertii*, 1:20	55.4	32.4	0
*A. seifertii*, undiluted	50.3	18.2	25
H_2_O	59.2	29.2	0

**Table 2 microorganisms-11-01133-t002:** Concentrations of acetate and ethanol, mM, and CO_2_, conventional units, % of the highest level in *A. seifertii* after 18-month storage) in the liquid phase of SHG-immobilized HOB preparations stored for 18 months.

Shg-Immobilized Culture	Storage in SHG, Months	Acetate, mM	Ethanol, mM	CO_2_, %
Control (SHG without cells)	0 and 18	225	213	0
*R. qingshengii*	6	91	184	63
	9	2	4	58
*A. seifertii*	9	32	153	67
	18	7	155	100
*P. extremoaustralis*	18	5	2	71

**Table 3 microorganisms-11-01133-t003:** Shares of *A. seifertii* and *R. qingshengii* cells obtained from the stationary and dormant cells capable of growth at high salinity.

Cell Type	NaCl 3%	NaCl 6%	NaCl 9%
*A. seifertii*			
Stationary	0.1%	0.0%	0.0%
DF	1.5%	0.0%	0.0%
*R. qingshengii*			
Stationary	100%	100% Growth on day 21	0.1%
DF	100%	100% Growth on day 21	10.0%

**Table 4 microorganisms-11-01133-t004:** Shares of *A. seifertii* and *R. qingshengii* cells obtained from the stationary and dormant cells capable of growth at high CuSO_4_ concentrations.

Cell Type	10^−4^ M CuSO_4_∙5H_2_O	10^−3^ M CuSO_4_∙5H_2_O	2 × 10^−3^ M CuSO_4_∙5H_2_O
*A. seifertii*			
Stationary	100%	4.9%	0.03%
DF	100%	17.5%	0.70%
*R. qingshengii*			
Stationary	100%	10%	0.00%
DF	100%	100%	1.00%

## Data Availability

Data is contained within the article.

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
