# Peer review of "Ways of Long-Term Survival of Hydrocarbon-Oxidizing Bacteria in a New Biocomposite Material—Silanol-Humate Gel"

_microorganisms, 2023, doi:10.3390/microorganisms11051133_

Round 1

Reviewer 1 Report

The manuscript can be accepted after addressing the following items: Aim of work is not clea and should be highlighted at the end of introduction part. In abstract, there some words are etallic written and underlined such as , the goal. check and amend. The source of each material should be mentioned in metarials section (2.) Figure 1, 2, and 4 should be redrawn with different colors to be more readable. The sacle bar for all microscopy images are messed!!!!. Standard error should be calculated and inserted with all values that presented in the figures.

The manuscript can be accepted after addressing the following items:
Aim of work is not clea and should be highlighted at the end of introduction part.
In abstract, there some words are etallic written and underlined such as , the goal. check and amend.
The source of each material should be mentioned in metarials section (2.)
Figure 1, 2, and 4 should be redrawn with different colors to be more readable.
The sacle bar for all microscopy images are messed!!!!.
Standard error should be calculated and inserted with all values that presented in the figures.

Author Response

Dear Reviewer!

Thank you very much for valuable notes and suggestions regarding out manuscript. We have changed the manuscript accordingly. Please find below our point-by-point reply.

On behalf of authors collective,

Dr Yury Nikolaev

  1. Aim of work is not clea and should be highlighted at the end of introduction

Answer: In the end of introduction part aim of the work is clearly written and marked by yellow.

  1. In abstract, there some words are etallic written and underlined such as, the goal. check and amend.

Answer: Mistakes were repaired and marked by yellow.

  1. The source of each material should be mentioned in metarials section

Answer: Appropriate changes are made and marked by yellow

  1. Figure 1, 2, and 4 should be redrawn with different colors to be more readable.

Answer: Figures 1, 2, and 4 have been redrawn with different colors and now are more readable.

  1. The scale bars for all microscopy images are missed!

Answer: We are sorry for such a rude omitting. It’s happened due to some computer discrepancies. Scale bars in all figs now are present.

  1. Standard error should be calculated and inserted with all values that presented in the figures.

Answer: Standard errors were calculated and now are present with all values in the figures.

Reviewer 2 Report

1. Title. Please change the title for “Long-term survival of hydrocarbon-oxidizing bacteria in a new biocomposite material — silanol-humate gel”.

Reasons to exclude the words “Mechanisms and forms”: mechanism is a system of causally interacting parts and processes that produce the phenomena; scientists have to explain the phenomena via the described mechanisms. The article presents not explanation but a nice description of the discovered phenomena / characteristics.

2. Abstract. The same comment for the abstract: please delete the sentence “The goal … electron microscopy” (Lines 16-18). Removing these lines will not cause any meaning loss to the article.

3. Introduction. Just recommendation to change the generalization “several orders of magnitude” (Line 63) for some specific data, for example “100-10,000”.

4. Materials and Methods. “P. aeruginosa” – Line 74 and through the text (Line 181 etc). The species Pseudomonas aeruginosa is well-known both by its degradation activity and as hazardous species. Thus, all strains of this species are forbidden for introduction into nature. Authors can use this species as a model but they cannot recommend it for introduction into natural ecosystem. Please mention this warning for some inexperienced readers which could accept the article as an approach to provide long-term survival for pathogenic bacteria.

5. Results. According to Lines 73-75, 4 strains were used in the experimental study. However, the data on preservation (Lines 181-184) are limited with 2 Gram-negative strains. What about 2 other strains?

The similar question for the Line 259: “Acetate was consumed completely by all three HOB strains…” – what about the 4th strain?

Line 270: please highlight the Latin name in italic.

Author Response

Dear Reviewer!

Thank you very much for valuable notes and suggestions regarding out manuscript. We have changed the manuscript accordingly. Please find below our point-by-point reply.

On behalf of authors collective,

Dr Yury Nikolaev

  1. Title. Please change the title for “Long-term survival of hydrocarbon-oxidizing bacteria in a new biocomposite material — silanol-humate gel”.

Reasons to exclude the words “Mechanisms and forms”: mechanism is a system of causally interacting parts and processes that produce the phenomena; scientists have to explain the phenomena via the described mechanisms. The article presents not explanation but a nice description of the discovered phenomena / characteristics.

Answer: I agree with reasons of respectable Reviewer and I removed “Mechanisms and forms” from the title. Instead I wrote “ways of…”, as actually the goal of the whole work/research was to describe ways and forms of very long survival. Phenomenon of long-term survival itself was described earlier.

  1. Abstract. The same comment for the abstract: please delete the sentence “The goal … electron microscopy” (Lines 16-18). Removing these lines will not cause any meaning loss to the article.

Answer: As I understand, sections “goal of the work” and “methods” are obligatory and are demanded by MDPI, so this sentence must not be removed. I have changed it in accordance with Reviewer’s recommendation.

  1. Introduction. Just recommendation to change the generalization “several orders of magnitude” (Line 63) for some specific data, for example “100-10,000”.

Answer: Repaired as recommended and highlighted by yellow.  

  1. Materials and Methods. “P. aeruginosa” – Line 74 and through the text (Line 181 etc). The species Pseudomonas aeruginosais well-known both by its degradation activity and as hazardous species. Thus, all strains of this species are forbidden for introduction into nature. Authors can use this species as a model but they cannot recommend it for introduction into natural ecosystem. Please mention this warning for some inexperienced readers which could accept the article as an approach to provide long-term survival for pathogenic bacteria.

Answer: Thank you very much for rational remark. In the end of 1st paragraph of Materials and Methods a sentence regarding P. aeruginosa was added and highlighted by yellow.

  1. Results. According to Lines 73-75, 4 strains were used in the experimental study. However, the data on preservation (Lines 181-184) are limited with 2 Gram-negative strains. What about 2 other strains?

Answer: Thank you for being good observer! Two other strains were just forgotten, we added them.

The similar question for the Line 259: “Acetate was consumed completely by all three HOB strains…” – what about the 4th strain?

Answer: Here it is true… In this specific experiment we investigated only three bacteria.

  1. Line 270: please highlight the Latin name in italic.

Answer: Done.